

# Association between receiving the *Aksi Bergizi* Social Behavioral Change Communication (SBCC) intervention and hygiene behaviors among secondary school students in Padang, Indonesia

Ricvan Dana Nindrea[1,2] and Wit Wichaidit[2]

[1] Department of Medicine, Faculty of Medicine, Universitas Negeri Padang, Bukittinggi, Indonesia
[2] Department of Epidemiology, Faculty of Medicine, Prince of Songkla University, Hat Yai, Thailand

## ABSTRACT

**Background**. The Government of Indonesia and UNICEF introduced the *Aksi Bergizi* Social Behavioral Change Communication (SBCC) intervention to promote healthy eating and hygiene behaviors among adolescents. However, no systematic assessment of the program's effect has been made. This study aims to assess the association between exposure to the *Aksi Bergizi* nutrition promotion program and hand, oral, and nail hygiene behaviors among secondary school students in Padang, Indonesia.

**Methods**. We conducted a school-based cross-sectional study in Padang Municipality, Indonesia, collecting data from 253 students attending *Aksi Bergizi* target schools and 253 students from non-target schools using a self-administered questionnaire. We compared hygiene behaviors between students in the two groups using descriptive statistics and multivariable logistic regression with adjustment for demographic and socioeconomic characteristics.

**Results**. All students reported brushing their teeth at least twice per day, so there was no observable difference regarding oral hygiene. However, we found that students in target schools were significantly more likely than those in non-target schools to always use soap when washing their hands both before eating (75% *vs.* 21%; Adjusted odds ratio (OR) = 6.03; 95% confidence interval (CI) [3.96–9.19]) and after using the toilet (74% *vs.* 21%; Adjusted OR = 5.74, 95% CI [3.78–8.72]). However, there was no statistically significant difference with regard to nail hygiene, *i.e.*, cutting nails at least once per week.

**Conclusion**. We found differences between target and non-target schools regarding self-reported handwashing but no differences in nail-clipping. The findings of this study have implications for stakeholders in infectious diseases and nutrition. Future studies should consider ways to reduce social desirability bias and increase the generalizability of the study findings.

Corresponding author
Wit Wichaidit, wit.w@psu.ac.th

## INTRODUCTION

Among adolescents, hygienic behaviours play a critical role in maintaining overall health and preventing the spread of infectious diseases (*Jatrana et al., 2021*; *Singh et al., 2023*). Hygiene refers to the practice of maintaining the cleanliness of body parts. The most common type of hygiene studied in public health is hand hygiene, which includes hand washing with soap and water as well as the use of hand rubs (*e.g.*, hand sanitizers) (*UNICEF, 2022*). Hand hygiene is universally recognized as a fundamental practice for preventing gastrointestinal and respiratory infections (*Aiello et al., 2008*). Oral hygiene refers to the practices that maintain the cleanliness and health of the mouth, teeth, and gums. This includes regular brushing and flossing of teeth, using mouthwash, and visiting the dentist for check-ups and cleanings (*NIDCR, 2024*). Poor oral hygiene after meals can lead to dental caries, gum diseases, and other health issues (*Duangthip & Chu, 2020*; *Santhosh et al., 2023*). Nail hygiene refers to the practices and behaviors that keep the fingernails and toenails clean, healthy, and free from harmful microorganisms (*CDC, 2024*). This includes regularly trimming nails, cleaning under them, and avoiding habits like biting nails or using unclean tools that can introduce germs or cause damage (*CDC, 2024*; *Wu & Lipner, 2020*). However, despite their importance, oral and nail hygiene practices are often overlooked as components of hygiene (*Wu & Lipner, 2020*), even among adolescents (*Su et al., 2024*). To promote hygiene and other health behaviors, various actors in health have applied Social Behavioral Change Communication (SBCC) tools. SBCC refers to the ''systematic application of theory-based, research-driven communication strategies'' (*USAID, 2019*) to influence social conditions and modify individual behaviors (*McKee, Becker-Benton & Bockh, 2014*).

Indonesia is a middle-income country in Southeast Asia where eating with hands is a common cultural practice, making hand hygiene promotion essential for reducing the risk of infectious diseases (*Han et al., 2020*). Health promotion programs in Indonesia that focus on adolescent nutrition may also include hygiene promotion. Despite this inclusion, hygiene behaviors are not as thoroughly described in the evaluation of these programs as dietary behaviors (*Loveday et al., 2014*). The Indonesian government, in collaboration with UNICEF, has implemented the *Aksi Bergizi* (''Nutrition Action'') program in 2022 in selected secondary schools nationwide. The program aimed to promote proper nutrition among adolescents based on SBCC principles (*Ministry of Health of Indonesia, 2019b*; *Ministry of Health of Indonesia, 2019a*; *UNICEF, 2019*), but also included hand, oral, and nail hygiene promotion activities. Information regarding hand, oral, and nail hygienes among students in schools that did and did not receive the *Aksi Bergizi* program can provide basic information and empirical evidence for stakeholders in adolescent health and infectious diseases. Therefore, the objective of this study is to assess the association between exposure to the *Aksi Bergizi* nutrition promotion program and hand, oral, nail hygiene behaviors among secondary school students in Padang, Indonesia.

## MATERIALS & METHODS

### Study design and setting

We conducted a school-based cross-sectional study in Padang Municipality, West Sumatra Province, Indonesia.

### Study participants and sample size calculation

The study participants were secondary school students aged 12-15 years in Padang Municipality, West Sumatra Province, Indonesia. Sample size calculation for the study was based on the main objective of comparing dietary habits between students of the target *vs.* non-target schools. We hypothesized that 59% of students in schools that received the *Aksi Bergizi* program had good knowledge of dietary habits (p1 = 0.59) compared to 42% of students in schools without the program (p2 = 0.42) (*Davis et al., 2009*; *Ndagire, Muyonga & Nakimbugwe, 2019*). With a 95% confidence level and 80% power, we calculated the sample size for comparing two proportions to be 253 students per group, resulting in a total of 506 students.

### Exposure: *Aksi Bergizi* program activities

The *Aksi Bergizi* nutrition promotion program included a series of activities based on SBCC for nutritional education and health promotion. The health promotion component included hand, oral, and nail hygiene promotion. The program mandated teachers from the school or healthcare staff from a local *puskesmas* (health centers) with demonstrating proper handwashing techniques and educating the students on the importance of regular brushing for oral health and the need for regular nail care to prevent infections. Teachers and healthcare staff also provided visual aids and distributed hygiene kits containing soap, toothpaste, toothbrushes, and nail clippers to encourage students to behavioral adoption and maintenance (*Ministry of Health of Indonesia, 2019b*; *Ministry of Health of Indonesia, 2019a*; *UNICEF, 2019*). The activities were held on the first week of every month. At the time of data collection, the Aksi Bergizi activities had been ongoing at target schools for approximately 18 months.

### Outcomes: self-reported hygiene behaviors

We developed the hygiene behavior questions based on the *Aksi Bergizi* modules. The questionnaire included six questions across three domains: hand hygiene, oral hygiene, and nail hygiene, focusing on activities within the past 30 days. The questionnaire included three questions on hand hygiene: "*How often did you wash your hands before eating?*", "*How often did you wash your hands after using the toilet or latrine?*", and "*How often did you use soap when washing your hands?*". Answer choices ranged from "*Never*" to "*Always*". The questionnaire also included two questions on oral hygiene: "*How many times per day did you usually clean or brush your teeth?*" and "*How often did you use toothpaste that contains fluoride when you cleaned or brushed your teeth?*". Answer choices ranged from "*Never*" to "*Every day, more than twice per day*". The questionnaire included one question on nail hygiene: "*How often did you clip your nails?*" and the answer choices ranged from "*Never*" to "*One time per week*".

## Study instrument

Our study instrument was a self-administered paper and pen questionnaire. We designed the questionnaire in English and Bahasa Indonesia by adapting existing instruments. We conducted a pilot-test of the study instrument in Solok Regency, which is one of the regencies/cities in West Sumatra Province. Further revision of the questionnaire was carried out through an iterative process. We also assessed the validity and reliability of the instruments during this period. The questions we used underwent validity and reliability testing, resulting in a Cronbach's alpha value of >0.7. The final questionnaire contained 9 sections, including: (A) Demographic characteristics; (B) Exposure (Aksi Bergizi nutrition promotion program); (C) Behavioral drivers for dietary and health; (D) Dietary habits; (E) Personal Hygiene; (F) Perception of Non Communicable Diseases (NCDs); (G) Physical Activity; (H) Alcohol, Tobacco, and Substance Use, and; (I) Reproductive Health and Prevention of Sexually Transmitted Diseases (STDs). Each questionnaire took approximately 20 min to complete. The English translation of the instrument can be found in the Supplementary Information section.

## Sampling methodology

The secondary school database was obtained from the Padang Municipality Education Office, West Sumatra Province, Indonesia. The sampling technique used in this study was multistage stratified clustered sampling. In the first stage, the secondary school database was obtained from the Padang Municipality Education Office, West Sumatra Province, Indonesia. A stratum was created based on secondary schools being targeted and non-targeted for the *Aksi Bergizi* program. Two schools were then randomly selected from each group. There were four secondary schools (all public) that were targeted for the *Aksi Bergizi* program. Among the four schools, only two schools were in the junior high school level, and the investigators selected both schools for the study. There were 39 secondary schools that were not targeted for the program, and the investigators selected two junior high schools with travel time of at least 45 min from the target schools in order to avoid the spill-over effect. In each school, we performed a stratified random sampling of classrooms to select two classrooms per grade level using the list of classrooms provided by the school. To avoid any potential repercussions, we hereby refrain from disclosing the names of the schools.

## Data collection

We scheduled an appointment with the selected secondary schools to determine a feasible time and date for data collection in the selected classrooms. We then asked the principal or the teacher in charge to introduce the investigators to the students in the classrooms. The investigators informed the students about the study and requested their verbal informed consent. We requested and obtained a waiver of written informed consent and parental consent from our Institutional Review Boards (IRBs) in order to help us reassume the participants of their confidentiality. When we requested the waivers, we explained to the IRB that the study questionnaire contained sensitive issues, and that requesting parental permission might prevent participants with the outcomes of interest from participation

and potentially introduce selection bias. The information and consent processes were conducted in a group setting within the selected classes, but we allowed for individual questions and answers during the recruitment process. After the students expressed verbal consent, we directly distributed the study questionnaires to the students and asked them to start filling out the questionnaires. We also ensured that no teacher was present in the classroom at the time. At the end of the data collection period, the students placed their questionnaire in an opaque envelope provided by the investigators, and placed the envelope in a locked box in front of the classroom or at a designated location. We collected data from 9 November 2023 to 20 December 2023.

### Data management

We opened the secured box in a private location and performed data entry on the paper questionnaires using the KoboToolbox platform, which uploaded the entered data to a password-protected server. For each questionnaire, two investigators performed data entry separately, and the principal investigator (RDN) checked for discrepancies between the two versions with regard to the unique identification number (ID) and other values, checked the original questionnaire, and made corrections accordingly.

### Data analysis

We used descriptive statistics to describe the characteristics of the study participants. For the association between receiving *Aksi Bergizi* intervention and hygiene behavior, we used bivariate descriptive analyses as well as bivariable and multivariable logistic regression analyses. In multivariable analyses, we adjusted for the student's age and sex as well as the parents' education as proxies for family socioeconomic status, based on the findings of previous studies (*Jatrana et al., 2021*; *Wichaidit et al., 2019*). We did not include missing values in our analyses. We conducted all analyses at 95% level of confidence.

### Ethical considerations

This study received ethical approval from the ethics committee of the Faculty of Medicine at Prince Songkla University (Approval No. REC.66-248-18-2).

## RESULTS

A total of 253 students from the *Aksi Bergizi* target schools and 253 students from the *Aksi Bergizi* non-target schools participated in our study (response rate = 100%). There were significant socio-demographic differences between participants target and non-target schools regarding age distribution, the education level of the participants' father, and household monthly income (Table 1). Students in target schools self-reported higher adherence to hand hygiene at key moments for hand hygiene (before eating, after using the toilet or latrine) than students at non-target schools (Table 2). Students from target schools were also significantly more likely to report adherences to oral hygiene and nail hygiene than students from non-target schools.

Students at *Aksi Bergizi* target schools were significantly more likely than students at the non-target schools to self-reported handwashing with soap at all times *vs.* the otherwise

**Table 1  Characteristics of the study participants in *Aksi Bergizi* target and non-target schools.**

| Characteristic | *Aksi Bergizi* nutrition promotion program | | P-value |
|---|---|---|---|
| | Target (*n* = 253) | Non target (*n* = 253) | |
| **Sex** | | | |
| Male | 106 (41.9%) | 111 (43.9%) | 0.719 |
| Female | 147 (58.1%) | 142 (56.1%) | |
| **Age** | | | |
| 12 years | 12 (4.7%) | 22 (8.7%) | 0.036 |
| 13 years | 72 (28.5%) | 90 (35.6%) | |
| 14 years | 115 (45.5%) | 88 (34.8%) | |
| 15 years | 54 (21.3%) | 53 (20.9%) | |
| **Ethnicity** | | | |
| Minangnese | 146 (57.7%) | 167 (66.0%) | <0.001 |
| Javanese | 35 (13.8%) | 57 (22.5%) | |
| Bataknese | 8 (3.2%) | 0 (0%) | |
| Sundanese | 5 (2.0%) | 0 (0%) | |
| Others | 59 (23.3%) | 29 (11.5%) | |
| **Father's occupation** | | | |
| Civil servant/state enterprise | 54 (21.3%) | 74 (29.2%) | <0.001 |
| Private sector employee | 92 (36.4%) | 79 (31.2%) | |
| Small-scale vendors/service providers | 42 (16.6%) | 30 (11.9%) | |
| Business owner/entrepreneur | 26 (10.3%) | 10 (4.0%) | |
| Laborer/manual workers | 18 (7.1%) | 60 (23.7%) | |
| Agriculture/ fishery | 19 (7.5%) | 0 (0%) | |
| Independent professions (*e.g.*, lawyers, architects) | 2 (0.8%) | 0 (0%) | |
| **Father's education** | | | |
| Junior high school | 38 (15.0%) | 65 (25.7%) | <0.001 |
| Senior high school | 86 (34.0%) | 93 (36.8%) | |
| Vocational certificate | 1 (0.4%) | 0 (0%) | |
| Associate's degree | 70 (27.7%) | 26 (10.3%) | |
| Bachelor's degree | 51 (20.2%) | 69 (27.3%) | |
| Higher than bachelor's degree | 7 (2.8%) | 0 (0%) | |
| **Mother's occupation** | | | |
| Housewife | 173 (68.4%) | 199 (78.7%) | 0.036 |
| Civil servant/ state enterprise | 15 (5.9%) | 12 (4.7%) | |
| Private sector employee | 22 (8.7%) | 12 (4.7%) | |
| Small-scale vendors/ service providers | 39 (15.4%) | 30 (11.9%) | |
| Business owner/ entrepreneur | 4 (1.6%) | 0 (0%) | |
| **Mother's education** | | | |
| Primary school | 2 (0.8%) | 0 (0%) | 0.031 |
| Junior high school | 45 (17.8%) | 63 (24.9%) | |
| Senior high school | 101 (39.9%) | 101 (39.9%) | |

**Table 1** (*continued*)

| Characteristic | *Aksi Bergizi* nutrition promotion program | | P-value |
| --- | --- | --- | --- |
| | Target (*n* = 253) | Non target (*n* = 253) | |
| Vocational certificate | 4 (1.6%) | 10 (4.0%) | |
| Associate's degree | 80 (31.6%) | 69 (27.3%) | |
| Bachelor's degree | 21 (8.3%) | 10 (4.0%) | |
| **Household monthly income** | | | |
| <1,000,000 IDR | 0 (0%) | 0 (0%) | <0.001 |
| 1,000,000 to 2,000,000 IDR | 31 (12.3%) | 66 (26.1%) | |
| 2,000,001 to 3,000,000 IDR | 61 (24.1%) | 77 (30.4%) | |
| 3,000,001 to 4,000,000 IDR | 95 (37.5%) | 72 (28.5%) | |
| 4,000,001 to 5,000,000 IDR | 47 (18.6%) | 33 (13.0%) | |
| 5,000,001 to 6,000,000 IDR | 19 (7.5%) | 5 (2.0%) | |
| **Religion** | | | |
| Islam | 240 (94.9%) | 240 (94.9%) | 0.999 |
| Christianity | 13 (5.1%) | 13 (5.1%) | |
| **Body mass index (BMI)** | | | |
| Underweight (BMI < 18.5 kg/m$^2$) | 3 (1.2%) | 2 (0.8%) | 0.145 |
| Normal (BMI 18.5–22.9 kg/m$^2$) | 245 (96.8%) | 238 (94.1%) | |
| Overweight (23–24.9 kg/m$^2$) | 5 (2.0%) | 13 (5.1%) | |
| Obesity ($\geq$25 kg/m$^2$) | 0 (0%) | 0 (0%) | |
| **Main sources of information about health behaviors** | | | |
| Television advertisements | 63 (24.9%) | 41 (16.2%) | 0.003 |
| Family and/or friends | 131 (51.8%) | 169 (66.8%) | |
| Social media platforms (*e.g.*, Facebook, X, Instagram, *etc.*) | 59 (23.3%) | 43 (17.0%) | |

before eating (75% *vs.* 21%; Adjusted OR = 6.03; 95% CI [3.96–9.19]) (Table 3) and after using the toilet (74% *vs.* 21%; Adjusted OR = 5.74, 95% CI [3.78–8.72]). However, there was no statistically significant association between school group and clipping nails.

## DISCUSSION

In this school-based cross-sectional study, we compared self-reported hygiene behaviors among students from schools that received the *Aksi Bergizi* nutrition promotion program with hygiene-related activities *vs.* students from schools that did not receive the program. We found universal compliance to oral hygiene. We also found significant differences regarding the probability of handwashing with soap before eating and after toilet use, but no statistically significant difference regarding nail hygiene behaviour.

The findings of our study regarding hand hygiene align with these previous studies (*Jatrana et al., 2021*; *Rahman Zuthi et al., 2022*). The additional contribution of our study findings may be the data pertaining to oral hygiene and nail hygiene behaviors, which were less commonly studied. In that regard, all of the students in our study, regardless of school type, reported brushing teeth at least twice per day. Although the participants
**Table 2  Self-reported hygiene behaviors among students exposed to the *Aksi Bergizi* program and students not exposed to the program.**

| Personal hygiene | *Aksi Bergizi* nutrition promotion program | | P-value |
|---|---|---|---|
| | Target (*n* = 253) | Non target (*n* = 253) | |
| **Hand hygiene** | | | |
| **During the past 30 days, how often did you wash your hands before eating?** | | | <0.001 |
| Never | 0 (0%) | 0 (0%) | |
| Rarely | 0 (0%) | 0 (0%) | |
| Sometimes | 0 (0%) | 38 (15.0%) | |
| Most of the time | 80 (31.6%) | 161 (63.6%) | |
| Always | 173 (68.4%) | 54 (21.3%) | |
| **During the past 30 days, how often did you wash your hands after using the toilet or latrine?** | | | <0.001 |
| Never | 0 (0%) | 0 (0%) | |
| Rarely | 0 (0%) | 0 (0%) | |
| Sometimes | 1 (0.4%) | 5 (2.0%) | |
| Most of the time | 84 (33.3%) | 194 (76.7%) | |
| Always | 167 (66.3%) | 54 (21.3%) | |
| **During the past 30 days, how often did you use soap when washing your hands?** | | | <0.001 |
| Never | 0 (0%) | 0 (0%) | |
| Rarely | 0 (0%) | 30 (11.9%) | |
| Sometimes | 14 (5.5%) | 71 (28.2%) | |
| Most of the time | 80 (31.6%) | 97 (38.5%) | |
| Always | 159 (62.8%) | 54 (21.4%) | |
| **Oral hygiene** | | | |
| **During the past 30 days, how many times per day did you usually clean or brush your teeth?** | | | <0.001 |
| Never | 0 (0%) | 0 (0%) | |
| Not every day | 0 (0%) | 0 (0%) | |
| Every day, once per day | 0 (0%) | 0 (0%) | |
| Every day, twice per day | 201 (79.4%) | 237 (93.7%) | |
| Every day, more than twice per day | 52 (20.6%) | 16 (6.3%) | |
| **During the past 30 days, how often did you use a toothpaste that contains fluoride when you cleaned or brushed your teeth?** | | | <0.001 |
| Never | 0 (0%) | 0 (0%) | |
| Not every day | 0 (0%) | 0 (0%) | |
| Every day, once per day | 0 (0%) | 0 (0%) | |
| Every day, twice per day | 198 (78.3%) | 237 (93.7%) | |
| Every day, more than twice per day | 55 (21.7%) | 16 (6.3%) | |

**Table 2** (*continued*)

| Personal hygiene | *Aksi Bergizi* nutrition promotion program | | *P*-value |
|---|---|---|---|
| | Target (*n* = 253) | Non target (*n* = 253) | |
| **Nail hygiene** | | | |
| **During the past 30 days, how often did you clip your nails?** | | | 0.001 |
| Never | 0 (0%) | 0 (0%) | |
| One time per week | 207 (81.8%) | 217 (85.8%) | |
| One time per two weeks | 41 (16.2%) | 20 (7.9%) | |
| One time per three weeks | 5 (2.0%) | 16 (6.3%) | |
| One time per four weeks or did not clip | 0 (0%) | 0 (0%) | |

**Table 3** Association between attending *Aksi Bergizi* target *vs.* non-target schools and self-reported hygiene behaviors.

| Behavior | No outcome | Outcome | Unadjusted OR (95% CI) | Adjusted OR (95% CI)[*] |
|---|---|---|---|---|
| **Handwashing before eating** | **Does not always wash hands or does not always use soap during handwashing** | **Always wash hands and always uses soap** | | |
| *Aksi Bergizi* non-target schools | 199 (78.7%) | 54 (21.3%) | *Reference* | *Reference* |
| *Aksi Bergizi* target schools | 94 (32.1%) | 159 (74.6%) | **6.23 (4.20, 9.25)** | **6.03 (3.96, 9.19)** |
| **Handwashing after toilet use** | **Does not always wash hands or does not always use soap during handwashing** | **Always wash hands and always uses soap** | | |
| *Aksi Bergizi* non-target schools | 199 (78.7%) | 54 (21.3%) | *Reference* | *Reference* |
| *Aksi Bergizi* target schools | 96 (32.5%) | 156 (74.3%) | **5.99 (4.04, 8.88)** | **5.74 (3.78, 8.72)** |
| **Nail hygiene** | **Cut nails less than once per week** | **Cut nails once per week** | | |
| *Aksi Bergizi* non-target schools | 36 (14.2%) | 217 (85.8%) | *Reference* | *Reference* |
| *Aksi Bergizi* target schools | 46 (18.2%) | 207 (81.8%) | 0.75 (0.46, 1.20) | 0.64 (0.38, 1.08) |

**Notes.**
[*]Adjusted for age and sex of the child, father's education, and mother's education
Bold numbers denote statistical significance at 95% level of confidence.

self-administered the questionnaire, they might have feared being overseen by their peers and thus were less likely to self-report socially undesirable behaviors, and social desirability is common in self-reporting oral hygiene practices (*AlGhamdi et al., 2020*; *Sanzone et al., 2013*). Future studies should consider alternative ways to measure oral hygiene that may be less prone to social desirability, such as asking participants to click on the link for the questionnaire when they are home and safe from being overseen by peers.

While the study highlighted significant disparities in hygiene behaviors, several important aspects remained unknown or understudied. Firstly, previous studies have found gaps btween self-reported *vs.* observed hygiene behaviors (*Hagiya et al., 2022*; *Seyed Nematian et al., 2017*). Future studies should consider including a structured observation of hand hygiene behavior, similar to that in a previous study (*Wichaidit et al., 2019*). Furthermore, hygiene is influenced by disgust and social norms (*Nizame et al., 2013*; *Wichaidit et al., 2019*). We did not include measurement questions for these behavioral drivers in our

study. Future studies should consider developing a theory of behavior change for the intervention, and collect data on potential drivers of hygiene behaviors and make pathway analyses according to the developed theory to achieve a more complete understanding of the observed differences.

The strength of our study was the high level of voluntary participation, which reduced selection bias from non-response. However, a number of limitations should be considered in the interpretation of the study findings. Firstly, the intervention was not randomized, and the cross-sectional design of the study precludes the ability to make inferences regarding program effectiveness. Secondly, hygiene behavior is deemed in many regions, including the study area, to be socially desirable. Thus, the influence of social desirability may be non-negligible in our study findings. Lastly, we conducted our study only among schools in Padang, West Sumatera. The findings of the study may not be generalizable to students and schools in other regions.

## CONCLUSIONS

We found that students in schools that received the *Aksi Bergizi* nutritional promotion program were more likely to self-report handwashing with soap before eating and after toilet use, but there was no significant difference regarding frequency of nail-clipping. Considering the role of hygiene in improving nutrition, the findings of this study may be relevant to stakeholders in both adolescent infectious diseases and adolescent nutrition. However, limitations regarding the cross-sectional design, social desirability, and limited generalizability should be considered in the interpretation of the study findings.

## ACKNOWLEDGEMENTS

We wish to thank all study participants and study school teachers and staff for their valuable time and assistance throughout this study. We also wish to thank our research assistants and data entry staff for their tireless efforts. We would like to thank Prof. Sawitri Assanangkornchai and Prof. Virasakdi Chongsuvivatwong for the conceptualization and design of the study, and Prof. Masrul Muchtar for the kind assistance with the data collection process in the study area. We would like to thank the Padang Municipality Education Office of the Republic of Indonesia for their support and permission during the conduct of this study. This study was part of RDN's thesis, which was completed in partial fulfillment of the requirements for a Master of Science (M.Sc) degree in Epidemiology.

### Funding

This study was supported by the TUYF Charitable Trust: Research Capacity through Education and Networking on Epidemiology in Asia, the Department of Epidemiology, Faculty of Medicine, Prince of Songkla University (Grant number 1/2022). The funders had no role in study design, data collection and analysis, decision to publish, or preparation of the manuscript.

## Grant Disclosures

The following grant information was disclosed by the authors:

TUYF Charitable Trust: Research Capacity through Education and Networking on Epidemiology in Asia.

Department of Epidemiology, Faculty of Medicine, Prince of Songkla University: 1/2022.

## Competing Interests

The authors declare there are no competing interests.

## Author Contributions

- Ricvan Dana Nindrea conceived and designed the experiments, performed the experiments, analyzed the data, prepared figures and/or tables, authored or reviewed drafts of the article, and approved the final draft.
- Wit Wichaidit conceived and designed the experiments, analyzed the data, prepared figures and/or tables, authored or reviewed drafts of the article, and approved the final draft.

## Human Ethics

The following information was supplied relating to ethical approvals (i.e., approving body and any reference numbers):

This study received ethical approval from the ethics committee of the Faculty of Medicine at Prince Songkla University (Approval No. REC.66-248-18-2).

## Data Availability

The raw data are available in the Supplementary File.

## Supplemental Information

Supplemental information for this article can be found online at http://dx.doi.org/10.7717/peerj.19256#supplemental-information.

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
