# Peer review of "Association between receiving the *Aksi Bergizi* Social Behavioral Change Communication (SBCC) intervention and hygiene behaviors among secondary school students in Padang, Indonesia"

_PeerJ, doi:10.7717/peerj.19256_

## Round 0.1 · original submission · Minor Revisions

Please address the comments of the reviewers, and be aware that R2 has also included additional comments in an appended PDF

Reviewer 1 ·

Basic reporting

.

Experimental design

.

Validity of the findings

.

Additional comments

1. Congratulations to the Authors for highlighting a study which is an
important consideration in maintaining a good nutritional status.
Although a common routine practice, it is often overlooked. Thus, this
study was interesting in that perspective.

2. The 1st paragraph in the Introduction should describe what the Aksi
Bergizi SBCC tool is used for. There is a mix of Past and present
tense, esp in the 2nd para of the Introduction.

3. Sentence construct can be improved or rephrased - Lines 61-63,
instead of including, can be written as even among adolescents. To
rephrase lines 64-65. Line 99- is it tasked or targeted?

4. Lines 213-215 need clarity.
5. Line 228- can be written as voluntary participation reduced bias,
instead of lack of refusal.
6. Under Methods, more clarity needs to be given for the basis of
Hypothesis. What did the Study instrument contain? These need to be
elaborated.
7. Under Data collection, there is no mention of a written consent
from the students, or any mention of parental consent.
8. Conclusion can be rewritten to emphasise the role of these
practices in maintaining a good nutritional status.

Reviewer 2 ·

Basic reporting

The conclusion in the abstract is revised. Adjust it to the research results. Use the existing conclusion as a suggestion.

Experimental design

Add an explanation of how to determine target schools

Validity of the findings

No Comment

Additional comments

No Comment

Annotated reviews are not available for download in order to protect the identity of reviewers who chose to remain anonymous.

---

## Round 0.2 · accepted · Accept

I have reviewed your point-by-point responses and the revised manuscript. Your changes have adequately addressed all the concerns raised by the reviewers. Given the minor nature of the previous revision and the quality of your responses, I am pleased to inform you that your manuscript has been accepted for publication.

Reviewer 1 ·

Basic reporting

.

Experimental design

.

Validity of the findings

.

Additional comments

Since the authors have clarified all the doubts and made corrections, the article can be considered for publication.